# Impact of the New Coronavirus Infection on the Immune System of Children and Adolescents in the Region of the Russian Federation

**DOI:** 10.3390/ijerph192013669

**Published:** 2022-10-21

**Authors:** Sergey Kostarev, Oksana Komyagina, Rustam Fayzrakhmanov, Daniel Kurushin, Natalya Tatarnikova, Oksana Novikova (Kochetova), Tatyana Sereda

**Affiliations:** 1Perm National Research Polytechnic University, 29, Komsomolski Avenue, Perm 614990, Russia; 2Perm State Agro-Technological University Named after Academician D N Pryanishnikov, 23, Petropavlovskaja St., Perm 614990, Russia; 3Perm Institute of the FPS of Russia, 125, Karpinskogo St., Perm 614012, Russia; 4Medical Institution “Philosophy of Beauty and Health”, 64, KIM St., Perm 614990, Russia

**Keywords:** immunograms, SARS-CoV-2, immunology

## Abstract

The emergence of COVID-19 (SARS-CoV-2) has presented public health professionals with new challenges in the diagnosis of the disease and treatment of patients. Nowadays, the epidemiology, clinical features, prevention and treatment of the disease are studied poorly due to continuous mutation of the pathogen. One of the consequences of the new coronavirus infection could be changes in the immune system of the human population. A detailed analysis of the immunological status of different racial groups under the influence of the new coronavirus infection is currently studied insufficiently, making this work of particular relevance. There is also a reluctance among some Russian residents to be vaccinated, including the population of Perm Krai, due to a lack of research on possible deviations in cellular immunity due to SARS-CoV-2 vaccination. At the start of the third wave caused by the new coronavirus infection, only 40% of the Russian population had been vaccinated, which was insufficient to acquire collective immunity. In the autumn of 2021, a QR code measure was introduced for vaccinated residents, which resulted in exceeding the necessary barrier for acquiring collective immunity. Due to the high growth and severity of the disease, we analysed the immunograms of children and adolescents, aged from 5 months to 17 years, in Perm Krai during the pandemic years 2020–2021. The patients’ immunological status results were divided into three categories. Laboratory diagnosis of the human immune system was carried out using serological and flow cytophotometric analyses. A total of 247 samples were analysed. The aim of this work was to investigate changes in the immune system of children and adolescents during the pandemic caused by the new coronavirus infection. The methodology was based on the analysis of immunograms, including biochemical studies, immune status and flow cytophotometric analysis. The immunograms were pre-sorted by IgA, IgM, IgG immunoglobulin status into four categories: absence of disease—*k*_1_ in which IgA, IgM, IgG immunoglobulin values were within the reference interval, active disease stage—*k*_2_ in which IgA, IgM immunoglobulins had gone beyond the reference interval, passive disease stage—*k*_3_ characterised by IgG and IgM immunoglobulin status, and patient recovery process—*k*_4_. In the immunograms, three immune status indicators were selected for further investigation: phagocytosis absolute value, phagocytic number and phagocytic index and five flow cytometry indices: leukocytes, lymphocytes, NK cells (CD16+CD56+), T helpers (CD3+CD4+) and CD4+/CD8+ immunoregulation index. A quantitative analysis of the deviations of these indicators from the reference intervals was performed in the three studied age groups of children and adolescents living in Perm Krai of the Russian Federation during the pandemic of 2020–2021.

## 1. Introduction

The foundations of immunology were established in the scientific school of the French scientist Louis Pasteur [1], which were further formulated by his apprentice, the Russian scientist and Nobel Prize winner Elya I. Mechnikoff [2]. Many review papers have been devoted to the study of the human immune system, but the possible deterioration in immune performance caused by the new coronavirus infection is of particular concern [3,4]. Coronaviruses are RNA+ viruses and belong to the order Nidoviralec, family *Coronaviridae*, which includes 2 subfamilies *Toroviridna* and *Coronaviridna* (genera *Alphavirus*, *Betavirus*, *Gammavirus*) [5,6]. Four strains of coronaviruses, including: HCoV-229E, -OC43, -NL63 and -HKU1 have been circulating continuously previously and currently and have caused acute respiratory viral disease [7,8]. On 11 February 2020, the World Health Organization determined the official name of the infection caused by the new coronavirus—COVID-19. The International Committee on Taxonomy of Viruses assigned an official name to the infectious agent—SARS-CoV-2. In Perm Krai, the first case of a new coronavirus infection (SARS-CoV-2) was detected in March 2020 in a man returning from abroad. Between 2020 and 2021, there were three waves of coronavirus infection, which had an impact on the immune parameters of the residents of Perm Krai [9]. The large range of issues and phenomena studied in relation to immunity has proved to be complex and varied. Immunology studies specific and nonspecific protective and adaptive mechanisms that determine and regulate the homeostasis of the body’s environment. It also studies humoral immunity factors (antibodies, bactericidal properties of sera), cell-tissue reactions (phagocytosis, cell reactivity) and general physiological processes that condition immunity. Much attention in immunology is given to the study of the nature and properties of antigens, i.e., substances that induce general immunological reactions (these include microbes, their toxins, proteins, polysaccharides and viruses). The peculiarities of the biochemical composition and structure of the virion, the uniqueness of their biology and their interactions with cells are reflected in the manifestation of protective reactions to viruses, both on the part of cells and on the part of the functions of the entire organism. Viruses are obligate intracellular parasites. In susceptible cells their main vital functions are deproteinization, induction of nucleic acids, enzyme synthesis, assembly and escape of mature virions outside the affected cell. Defence mechanisms target two forms of viral existence—extracellular (dormant) and intracellular (vegetative). Defence responses to the extracellular form of viruses include specific and non-specific ones, as well as cellular and humoral factors. The virus is an exogenous agent to which the body responds to by producing specific antibodies. Antibodies are formed not only to target the virion, but also to its components, e.g., to the outer antigen of macro viruses, and to the inner nucleoprotein [10]. Immune control cells include lymphocytes: B lymphocytes, which are responsible for humoral immunity and are relevant to antibody synthesis, and T lymphocytes, which are responsible for cellular immunity and are divided into T killers (killer cells), T helpers, which enhance immunological reactivity, and T suppressors, which weaken reactivity [11,12]. Perspectives on immune system modulation in the SARS-CoV-2 infection using India as an example are discussed in [13]. In the article [13], using the example of a country that cannot provide vaccination for the population (as it has a largely poor population), the article examines the role of the Indian immune system in investigating the mortality/infection ratio of COVID-19 in urban and rural areas. The relevance of the susceptibility of the child immune system to SARS-CoV-2 is explored in the article [14], which examines the specific traits of the development of the immune system of newborn children when they are infected with SARS-CoV-2. The prevalence of the new coronavirus infection among Russian children has been studied in [15], where the data on the new coronavirus infection among children is summarized, and indicators and features of the epidemiology and clinical findings of the new infection are systematized. The status of cellular immunity in children with recurrent respiratory disease has been described in [16]. The highest number of children with indicators outside the reference range (RR) of various T and B lymphocyte subpopulations were observed at the age of 3–6 years. In children, decreased levels of B lymphocytes (CD19+) and T lymphocytes (CD3+CD8+) were accompanied by increased levels of total lymphocytes (CD3+) and T helpers (CD3+CD4+). The most frequent variants of the combined disturbances of cellular immunity were revealed: a decrease in B lymphocytes and an increase in T cells and T helpers—in 40.2% (206 of 512 children); a decrease in B lymphocytes and a decrease in T cytotoxic cells—in 42% (215 of 512). Similar patterns were detected in children with recurrent respiratory diseases in the Mogilev and Minsk regions: positive Spearman correlation coefficients were between CD3+/CD4+ T lymphocytes and also between CD3+/CD8+ T lymphocytes, negative coefficients were between CD4+/CD8+ T lymphocytes [16]. The problem of differential diagnosis of immunity to SARS-CoV-2 was considered at a conference in Sochi in 2021 [17]. The formation of collective immunity to SARS-CoV-2 among the population of the Republic of Belarus was studied in [18]. The paper [18] analysed the results of a study that focused on 2675 people. Collective immunity was 8.7% among children aged 1–6 years (14.5%). No statistically significant differences in seroprevalence were found between men and women. In asymptomatic individuals with a positive PCR result, specific antibodies were detected in 21.7% of cases. In 93.4% of seropositive individuals, the infection was asymptomatic. Markers of long-term immunity have been studied in [19]. The review [19] discusses a study that evaluated the humoral and cellular immune response to the BNT162b2 (Pfizer-BioNTech) anti-COVID-19 vaccine in patients receiving methotrexate. The rate of antibody production was lower in patients receiving methotrexate, though the level of T cell response was similar in all the groups studied. At present, the change in cellular immunity under the influence of the new coronavirus infection has not yet been sufficiently studied, which makes some residents of Perm Krai afraid to be vaccinated. Immunograms of three age groups of children and adolescents in Perm Krai during the pandemic period 2020–2021 were studied. Variation in immunoglobulin A, M and G norms was also taken into account in the analysis of the immunograms. An approach based on the analysis of immunoglobulins in different mutations of the SARS-CoV-2 virus was discussed in [20]. The state of the immunoglobulins indicates the current state of the disease, i.e., by the mutual quantification of immunoglobulins A, M and G, a transient disease course was inferred and a deviation of the leucocytic blood count and phagocytic count was identified.

## 2. Materials and Methods

The study used the theory of immunogram interpretation in inflammatory processes [21,22]. The study of patients and the generation of data for completing immunograms were carried out by the medical institution “Philosophy of Beauty and Health” (Perm). Data collection sites for immunograms were located in Perm Krai, including the regional city of Perm (11 sites) and the district centres of Gubakha, Krasnokamsk and Solikamsk, which had one site each. During the pandemic of 2020–2021, 247 people were tested for the new SARS-CoV-2 coronavirus infection. All 247 samples were used in the study. The study was conducted in accordance with the Declaration of Helsinki and approved by the Ethics Committee of Perm State Agrarian-Technological University named after the academician D N Pryanishnikov. The studies were conducted prior to vaccination of the patients, as groups of children and adolescents under 17 years of age were investigated, and due to the fact that vaccines for children were tested in the Russian Federation during this period and no compulsory infant vaccination was required. The studies were based on serological and flow cytophotometric analysis using an “ILab Taurus” automated analyser. The material used was patient venous blood. During flow cytophotometric analysis, blood was stabilized with anticoagulant. The MultiTEST IMK Kit reagent with dyes was used to detect antibodies. Before starting the work, a lysing solution was prepared using the MultiTEST IMK Kit lysing solution with 450 µL of solution used per test tube. Specific antibodies were determined by ELISA. The test procedure for the antibody class was based on two steps of solid-phase immunoassay: the first step was the process of binding the antibodies contained in the sample analysed to monoclonal antibodies to human immunoglobulins immobilized on the inner surface of the well; the second step was the formation of a complex binding antibodies to SARS-CoV-2 with a conjugate of recombinant antigen SARS-CoV-2 with horseradish peroxidase. Incubation with tetramethylbenzidine solution resulted in staining of the solution in the wells containing the formed immune complexes. The intensity of staining was proportional to the concentration of detectable antibodies in the analysed sample. After stopping the reaction by adding a stop reagent, the results of the analysis were determined by the optical density of the solution in the wells of the plate. Biochemical serological tests were based on IgA, IgM and IgG immunoglobulin status. Immune status (ES) was determined based on the determination of phagocytic indices: absolute value of phagocytosis, phagocytic number, and phagocytic index. The immunograms in the flow cytometry analysis were: leukocytes, lymphocytes, NK cells (CD16+CD56+), T helpers (CD3+CD4+) and CD4+/CD8+ immunoregulatory index. Quantitative reference intervals used in the laboratory of the medical institution “Philosophy of Beauty and Health” were used in the analysis of the immunity indicators. The immunogram indicators were studied for four groups of patients residing in Perm Krai, depending on the mutual state of immunoglobulins IgA, IgM, IgG. The theory of systems analysis and differential calculus was used to develop the mathematical models.

## 3. Research Results and Discussion

### 3.1. Study of the Pandemic in Perm Krai of the Russian Federation Subsection

The three waves of the worsening epidemiological situation related to SARS-CoV-2 virus were observed in Perm Krai of the Russian Federation during the period of 2020–2021. The first wave occurred in the spring of 2020. The 1 March 2020 is considered to be the beginning of the epidemic in Perm Krai, as the first cases of coronavirus infection began to be recorded then. On 2 April 2020 Vladimir Putin issued a decree “On Measures to Ensure Sanitary and Epidemiological Well-Being of the Population in the Russian Federation in Connection with the Spread of the New Coronavirus Infection (COVID-19)”. From 2 April to the end of May 2020, strict self-isolation measures were taken. The second wave was in the autumn of 2020. A study of the human infectious safety model under the influence of SARS-CoV-2 in the territory of Perm Krai is described in [9]. Building a tool model for the study of the ecosystem “coronavirus—vector—human—environment” is described in [23]. The third wave occurred in the autumn of 2021 and affected the entire territory of Russia. Between 30 October and 7 November 2021, many businesses and all leisure centres were closed. At the end of 2021, a new strain of the virus, Omicron, characterised by higher replication activity, arrived in Perm Krai due to a mutation. The first case of the Omicron strain in Perm Krai was detected on 29 December in a citizen who arrived in Perm from Egypt in transit via Moscow. In March–April 2022, a new strain of Omicron-Steles was detected, and its destructive features also included upper respiratory tract disease. Numerous papers have been devoted to studying the virus’ destructive features and models associated with it [24,25,26,27,28,29]. The peak period of the pandemic in Perm Krai could be the autumn–winter period 2021–2022, with more than 3000 cases per day. As of 3 March 2022, the total number of infected people in Perm Krai was 164,517 with a population of 2,555,042 according to Rosstat (The Federal State Statistics Service in Russia).

### 3.2. Development of the Immunogram Model

Three groups of parameters characterising the overall human immune system were taken in order to form the immunogram transformation model. The state of the human immune system *C_Im_* depends on the state of the flow-through digital photometry indicators *C_P_*, on the state of the phagocytosis indicators *C_F_* and the state of the immunoglobulin system *C_Ig_* is also an important indicator, depending on which the four categories of immunograms will be formed. In order to formalise the elements of the system, we will introduce sets of elements (1):(1)$$P=\left\{p_{1},p_{2},\ldots ,p_{n}\right\}\;F=\left\{f_{1},f_{2},\ldots ,f_{k}\right\}\;Ig=\left\{IgA,IgM,IgG\right\}\;Im=\left\{k_{1},k_{2},k_{3},k_{4}\right\}$$

*P*—the set of flow cytometry elements;*F*—the set of phagocytosis elements;*Ig*—the set of immunoglobulin elements;*Im*—the set of immunogram categories:
*k*_1_—absence of disease (Immunoglobulins *IgA*, *IgM*, *IgG* are in the reference interval);*k*_2_—active stage of the disease (Immunoglobulins *IgA*, *IgM*, out of the reference interval);*k*_3_—passive stage of the disease (Immunoglobulins *IgG* and *IgM*, exceeded the reference interval);*k*_4_—patient’s recovery process (*IgG* immunoglobulin is out of the reference interval).

The elements of the system will be linked by Boolean relations *R* (*R1, R2, R3*) [30], showing the mutual influence of arrays *P*, *F* and *Ig* on the state of the human immune system *C_Im_*:*P R1 C_Im_, Ig R2 C_Im_, F R3 C_Im_*.(2)

Figure 1 shows the relationship between the elements of the system under the study. It is possible to decompose the Boolean relation *R1*, *R2*, *R3* into two subsets: the physiological state of the immune system (0) *R1*^0^, *R2*^0^, *R3*^0^ and the pathological state of the immune system (1) *R1*^1^, *R2*^1^, *R3*^1^.

The immune system, like any other biological system, functions under certain physiological conditions of the biological environment. In this case, the system (2), with the introduction of the state variable (*C*), is transformed as follows (3):*PR1*^0^ [*C1*, *Im*^0^], *C1 R1*^1^
*Im*^1^, *IgR2*^0^ [*C2*, *Im*^0^], *C2 R2*^1^
*Im*^1^, *FR3*^0^ [*C3*, *Im*^0^], *C3 R3*^1^
*Im*^1^.(3)

The physiological state of the immune system is influenced by; intrinsic properties of the immune system {*S*}, indicators of biochemical examination for immunoglobulins, immune status indicators, the role of phagocytic numbers, and flow cytometry indicators. We define the state of the immune system by the dependence
*C_Is_* = Δ{*S*} + ΔΘ(*Ig*) + ΔΘ(*P*) + ΔΘ(*F*),(4)
where Θ is the index of the physiological state of the immune system, ΔΘ(*P*) is the change in the index of the physiological state of the immune system from the flow cytometry parameters, ΔΘ(*F*) is the change in the index of physiological state from the phagocytosis parameters, ΔΘ(*Ig*) is the change in the index of physiological state from immunoglobulin parameters.

Depending on the set of indicators of the immune analysis system elements, a more precise picture of the human immune state may be obtained. It is possible to describe a simplified system of elements (1), which is used in the immunogram of the medical centre “Philosophy of Beauty and Health”, Perm. In order to simplify the description of the immunogram parameters, we have also set up a system of identifiers (Table 1).

The indicator of the physiological state of the immune system {*S*} depends on the person’s own features: age (*Ag*), sex (*G*) and the influence of side diseases (*D*)
$$\Delta \left\{S\right\}=\frac{\partial \Theta }{\partial S_{Ag}}\Delta S_{Ag}+\frac{\partial \Theta }{\partial S_{G}}\Delta S_{G}+\frac{\partial \Theta }{\partial S_{D}}\Delta S_{D}.$$

The immunoglobulin index ΔΘ(*Ig*) is characterised by the deviation of the parameters of the three immunoglobulins *IgA*, *IgG*, *IgM*
$$\Delta \Theta \left(Ig\right)=\frac{\partial \Theta }{\partial IgA}\Delta IgA+\frac{\partial \Theta }{\partial IgM}\Delta IgM+\frac{\partial H_{P}}{\partial IgG}\Delta IgG.$$

The flow cytometry index ΔΘ(*P*) depends on the state of leukocytes (*p*_1_), lymphocytes (*p*_2_), NK cells (CD16+CD56+) (*p*_3_), T helpers (CD3+CD4+) (*p*_4_) and CD4+/CD8+ immunoregulation index (*p*_5_)
$$\Delta \Theta \left(P\right)=\frac{\partial \Theta }{\partial p_{1}}\Delta p_{1}+\frac{\partial \Theta }{\partial p_{2}}\Delta p_{2}+\frac{\partial \Theta }{\partial p_{3}}\Delta p_{3}+\frac{\partial \Theta }{\partial p_{4}}\Delta p_{4}+\frac{\partial \Theta }{\partial p_{5}}\Delta p_{5}.$$

The phagocytosis index ΔΘ(*F*) is described by immune status parameters
$$\Delta \Theta (F)=\frac{\partial \Theta }{\partial f_{1}}\Delta f_{1}+\frac{\partial \Theta }{\partial f_{2}}\Delta f_{2}+\frac{\partial \Theta }{\partial f_{3}}\Delta f_{3}.$$

Further studies were conducted on the deviation of the studied immunogram parameters during the 2020–2021 pandemic in the territory of Perm Krai. The study involved sampling sites in Perm, Krasnokamsk, Gubakha and Solikamsk, Perm Krai.

### 3.3. Laboratory Investigation of Children Immunograms

The experiments were carried out at the medical institution “Philosophy of Beauty and Health”, Perm, in the period 2020–2021. Immunograms were selected for populations of children and adolescents from 0 to 17 years of age. In order to simplify the calculations and to improve the visibility of the results, three age groups with increasing intervals were formed, according to the Weber–Fechner law: 0 to 3 years, 4 to 9 years, and 10 to 17 years. The study of immune status parameters and immunograms were grouped according to deviations in IgA, IgM and IgG immunoglobulins from the reference interval. A total of 247 immunograms of the immune system status were examined, including: *k*_1_ was 132 pc (53.4%), *k*_2_ was 80 pc (32.4%), *k*_3_ was 17 pc (6.9%), *k*_4_ was 18 pc (7.3%) (Table 2).

Immune status and flow cytometry indicators from biochemical immunoglobulin tests were then analysed for immune system states *k*_1_, *k*_2_, *k*_3_ and *k*_4_ (Table 3, Table 4, Table 5 and Table 6). The symbol ↑ is used to indicate an excess of the reference interval (RI), the symbol ↓ is used to indicate a subsidence from the reference interval.

In order to improve the visual perception of the deviations in the immunogram parameters, graphs were made. Figure 2 shows graphs of the percentage deviation in immunogram parameters for the immune system state *k*_1_.

Figure 3 shows graphs of the percentage deviations in the immunograms for the immune system state *k*_2_ when immunoglobulins IgA, IgM are increased (more) or decreased (smaller).

Figure 4 shows graphs of percentage deviations in immunograms for the *k*_3_ immune system state when immunoglobulins IgG and IgM are increased (more) or decreased (smaller).

Figure 5 shows graphs of the percentage deviations in immunograms for the *k*_4_ immune system state when immunoglobulin IgG is increased (more) or decreased (smaller).

### 3.4. Analysis of Immune Status and Immunograms from Biochemical Tests for Immunoglobulins

#### 3.4.1. Immune Status *k*_1_

Excess of the phagocytic number (Figure 2 (*f*_2_)) peaked in the middle age group up to 28% and declined to 13% in the older age group. A large percentage of the phagocytic number subsidence was observed in the older age group, 63%, and it decreased to 25% in the younger age group.

Phagocytic index (*f*_3_) had both peaks of excess (18%) and subsidence (32%) of the normal level in the middle age group. The largest subsidence was in the adolescent group, 59%.

Leukocytes (*p*_1_) were in excess of 28% in the younger and 12% in the older age groups. The decrease in leukocytes was insignificant.

Lymphocytes (*p*_2_) were in excess of 50% in the younger group. The decrease in lymphocytes was insignificant.

There were no particular abnormalities in NK cells (CD16+CD56+) (*p*_3_).

T helpers (CD3+CD4+) (*p*_4_) and immunoregulatory index (CD4+/CD8+) (*p*_5_) were significantly excessive in the younger age group. T helpers had a subsidence of 24% in the middle age group.

#### 3.4.2. Immune Status *k*_2_

Concerning the deviation in the immunoglobulins IgA, IgM—active disease stage (immune system state *k*_2_) it was found that phagocytosis (*f*_1_) was in excess in the middle age group up to 16%, and in excess up to 58% in the older age group with decreased immunoglobulins. A peak decrease to 83% was observed in the middle age group when immunoglobulins were in excess.

Phagocytic number (*f*_2_) had a peak subsidence of 75% with increased immunoglobulins in the middle age group and an average excess of 15% in the older group.

Phagocytic index (*f*_3_) had two peaks of subsidence, reaching up to 65% with increased immunoglobulins in the younger and middle age groups. Leukocytes (*p*_1_) were severely in excess with decreased immunoglobulins in the older group.

Lymphocytes (*p*_2_) were downgraded to 38% in the older group and NK cells (CD16+CD56+) (*p*_3_) were also downgraded to 70% with low immunoglobulin levels.

T helper cells (CD3+CD4+) (*p*_4_) had a subsidence of 30% when immunoglobulins were low and in excess of 36% when immunoglobulins were high in the older group. The immunoregulatory index (CD4+/CD8+) (*p*_5_) was above 60% in the younger group and 30% in the older group with increased immunoglobulins. A decrease in the immunoregulation index (36%) was observed in the older group with lower immunoglobulin levels.

#### 3.4.3. Immune Status *k*_3_

When immunoglobulins IgG and IgM were outside the reference interval, the passive stage of the disease (immune system state *k*_3_) showed a large decrease (77%) in phagocytosis (*f*_1_) in the older group with decreased immunoglobulin levels. An increase in phagocytosis (33%) was recorded in the middle age group.

Phagocytic count (*f*_2_) was increased in the middle group to 70% and decreased to 80% in the older group with decreased immunoglobulin levels.

The phagocytic index (*f*_3_) also decreased with decreased immunoglobulins in the older group.

Leukocytes (*p*_1_) were largely excessive (77%) in the older group with decreased immunoglobulins.

There was no abnormality observed in lymphocytes (*p*_2_) and NK cells (CD16+CD56+) (*p*_3_).

T helper cells (CD3+CD4+) (*p*_4_) were in excess of 33% and 66% in the middle and senior groups with decreased immunoglobulin levels and in excess of 16% in the middle group with increased immunoglobulin levels.

The immunoregulatory index (CD4+/CD8+) (*p*_5_) had a decrease of 77% in the older group and an increase of 33% in the middle age group with decreased immunoglobulin levels.

#### 3.4.4. Immune Status *k*_4_

During the convalescence of the patient (immunoglobulin IgG out of the reference interval, immune system status *k*_4_) there was a decrease in phagocytosis (*f*_1_) in the younger group (75%) and the middle group (50%) with a decreased immunoglobulin level, and excessive phagocytosis of 25% in the younger group with an increased immunoglobulin level.

Phagocytic count (*f*_2_) had two peaks: a decrease in the middle group (37%) and an increase in the younger group (12%) with decreased immunoglobulin levels. The phagocytic index (*f*_3_) had both an increase (50%) and a decrease (25%) in the middle group with lower immunoglobulin levels. The older group had a decrease of up to 80% with increased immunoglobulins.

The white blood cell (*p*_1_) and lymphocyte counts were identical, with an excess of up to 37% in the middle group at the lowered immunoglobulin level.

NK cells (CD16+CD56+) (*p*_3_) had excessive (50%) and decreased levels (37%) in the middle age group with decreased immunoglobulin levels.

T helpers (CD3+CD4+) (*p*_4_) were in excess in the younger group with increased and decreased immunoglobulin levels.

No abnormalities were found in the CD4+/CD8+ immunoregulation index (*p*_5_).

## 4. Conclusions

The study on the transformation of the immune system of children and adolescents during SARS-CoV-2 coronavirus infection 2020–2021 was conducted in Perm Krai of the Russian Federation. The study was conducted for three age groups according to the three indicators of biochemical studies and immune status, as well as the five indicators of flow cytophotometric analysis. The immunoglobulins were divided into four groups, for which the immune status deviation was investigated, including: phagocytosis absolute value, phagocytic number and phagocytic index and deviation of flow cytophotometric analysis indicators including five parameters: leukocytes, lymphocytes, NK cells (CD16+CD56+), T helpers (CD3+CD4+) and immunoregulation index (CD4+/CD8+).

The results of the study showed that in the *k*_1_ category (in the absence of deviations of immunoglobulins IgA, IgM, IgG from the reference interval) exhibited the largest decreases from the reference interval in the group of immune status indicators of phagocytosis with an increase in deviation towards the older age groups.

The highest exceedances from the reference interval were observed in the younger age groups for T helper (CD3+CD4+) and immunoregulatory index (CD4+/CD8+), indicating overactive immunity.

Patients in the *k*_2_ category (those in the active disease stage) also showed the greatest decreases from the reference interval in the indicator: phagocytic group for children of the middle and younger age groups with increased immunoglobulin levels.

The greatest exceedance of the reference interval was observed for leukocytes in the older group with lower immunoglobulin levels and for the immunoregulation index (CD4+/CD8+) in the younger group with increased immunoglobulin levels.

In children in the passive disease stage (*k*_3_), there were decreases from the reference interval in the phagocytic group and the immunoregulation index (CD4+/CD8+) for the older group with decreased immunoglobulin levels.

The greatest exceedances from the reference interval were noted for the middle age group for phagocytic number with decreased immunoglobulin levels and for the older age group for leukocytes and T helper cells.

In the *k*_4_ condition (healing process of a patient), the total deviation of the basic immunogram parameters decreased, but a decrease in the absolute phagocytosis value in the younger age group with a decreased immunoglobulin level and of the phagocytic index in the older age group with an increased immunoglobulin level were noted.

In a small number of cases, both increased and decreased immunoglobulins were found under the same conditions, which will require further research.

Studies are currently ongoing and when sufficient material has been accumulated the parameters of the new immunograms will be analysed and extended to the adult population.

## Figures and Tables

**Figure 1 ijerph-19-13669-f001:**
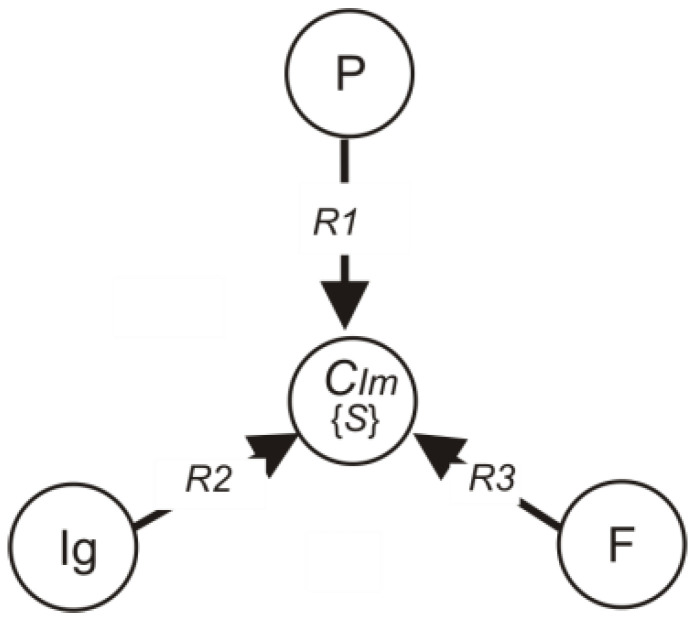
General structural scheme of connections between elements of the system: *R1*—effect of flow cytophotometry parameters; *R2*—effect of immunoglobulin parameters; *R3*—effect of phagocytosis parameters on the human immune system *CIm*.

**Figure 2 ijerph-19-13669-f002:**
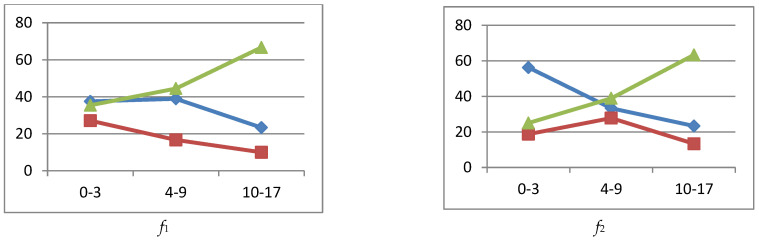
Graphs of deviations in immunogram indicators for the state of the immune system *k*_1_, 
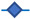
 reference interval, 
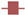
 exceeding the reference interval, 
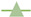
 subceeding the reference interval.

**Figure 3 ijerph-19-13669-f003:**
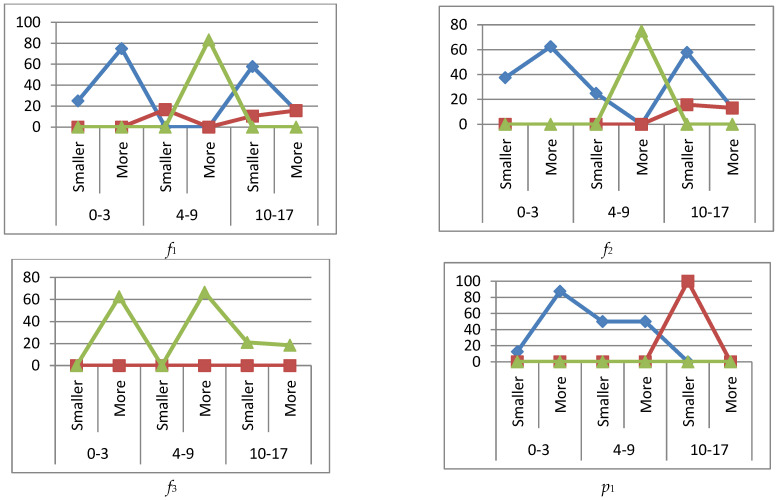
Graphs of deviations in immunogram indicators for the state of the immune system *k*_2_, 
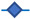
 reference interval, 
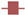
 more (exceeding the reference interval), 
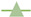
 smaller (subceeding the reference interval), IgA, IgM immunoglobulins are higher (more) or lower (smaller) if they are outside the reference interval.

**Figure 4 ijerph-19-13669-f004:**
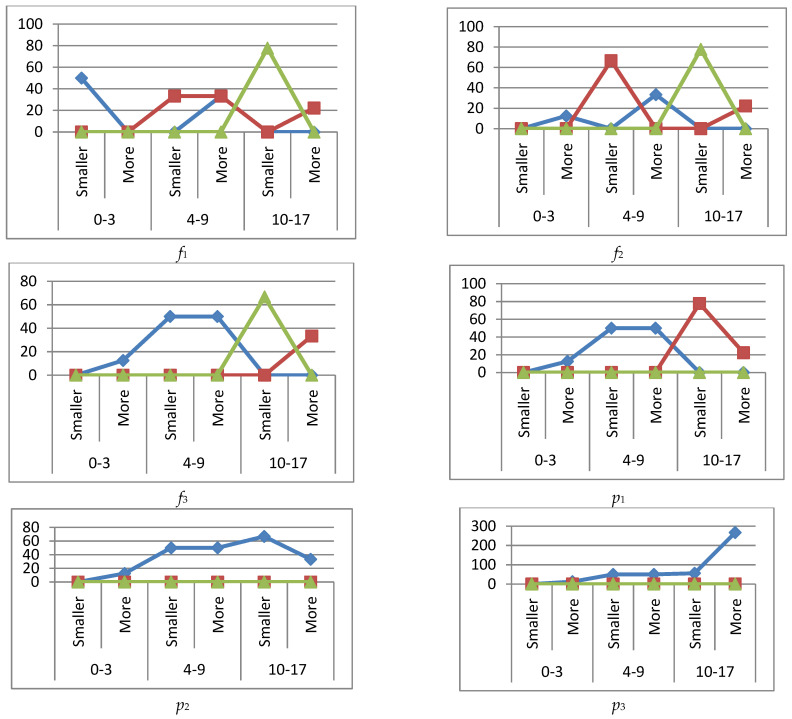
Graphs of deviations in immunogram indicators for the state of the immune system *k*_3_, 
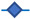
 reference interval, 
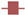
 more (exceeding the reference interval), 
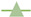
 smaller (subceeding the reference interval), when IgG, IgM immunoglobulins are higher (more) or lower (smaller) than the reference interval.

**Figure 5 ijerph-19-13669-f005:**
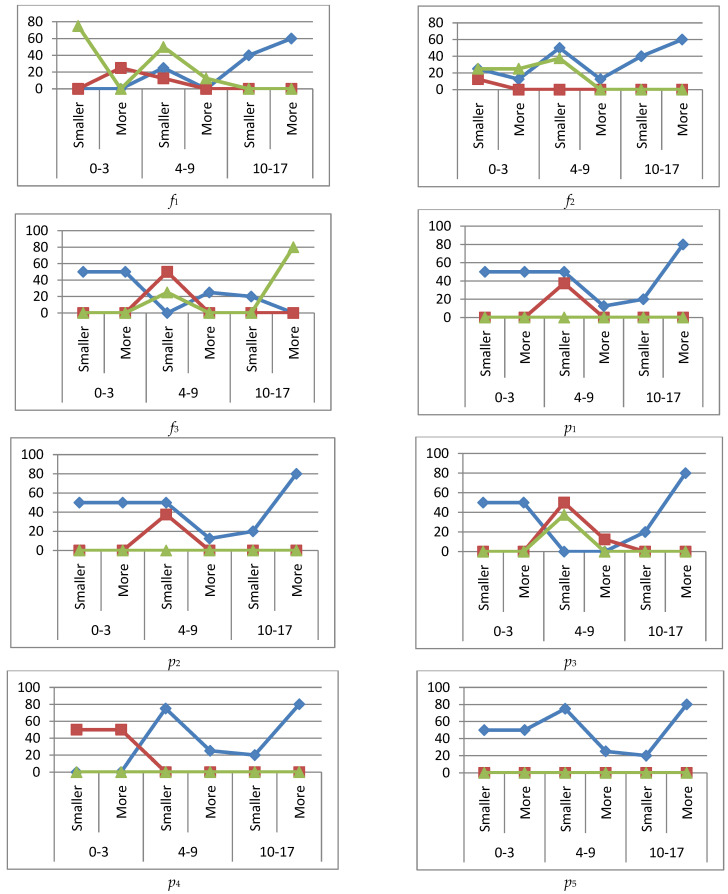
Graphs of deviations in immunogram indicators for the state of the immune system *k*_4_, 
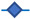
 reference interval, 
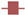
 more (exceeding the ref. interval), 
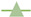
 smaller (subceeding the ref. interval), when immunoglobulin IgG is increased (more) or decreased (smaller) beyond the reference interval.

**Table 1 ijerph-19-13669-t001:** Coding of immunogram indicators.

Input Signals Indicators	Dimension	Mnemonics
Immunoglobulin A	g/L	A
Immunoglobulin M	g/L	M
Immunoglobulin G	g/L	G
Absolute value of phagocytosis	10^9^/L	*f* _1_
Phagocytic number		*f* _2_
Phagocytic index		*f* _3_
Leukocytes	10^9^/L	*p* _1_
Lymphocytes	10^9^/L	*p* _2_
NK cells (CD16+CD56+)	10^9^/L	*p* _3_
T helpers (CD3+CD4+)	10^9^/L	*p* _4_
(CD4+/CD8+) immunoregulation index		*p* _5_

**Table 2 ijerph-19-13669-t002:** Quantitative and percentage composition of patients by age category for immune system states *k*_1_, *k*_2_, *k*_3_, and *k*_4_ (the symbol ↑ is used to indicate an excess of the reference interval (RI), the symbol ↓ is used to indicate a subsidence of the reference interval).

Age Range, Years	The Number of Examined Samples						
	*k* _1_	*k* _2_		*k* _3_		*k* _4_	
		↓	↑	↓	↑	↓	↑
0–3	48	2	6	1	1	6	2
	72.7%	12.1%		3%		12.1%	
4–9	54	24	10	2	4	4	1
	70.1%	15.6%		7.8%		6.5%	
10–17	30	25	13	2	7	1	4
	36.5%	46.3%		10.9%		6.1%	
Total by subcategories	132	51	29	5	12	11	7
Total	132	80		17		18	
	53.4%	32.4%		6.9%		7.3%	

**Table 3 ijerph-19-13669-t003:** Immunogram analysis for the state of the immune system *k*_1_ (The symbol ↑ is used to indicate an excess of the reference interval (RI), the symbol ↓ is used to indicate a subsidence from the reference interval).

Indicator	Condition	Age Range, Years					
		0–3		4–9		10–17	
		RI	%	RI	%	RI	%
	RI	18	37.50	21	38.89	7	23.33
*f* _1_	↑	13	27.08	9	16.67	3	10.00
	↓	17	35.42	24	44.44	20	66.67
	RI	27	56.25	18	33.33	7	23.33
*f* _2_	↑	9	18.75	15	27.78	4	13.33
	↓	12	25.00	21	38.89	19	63.33
	RI	25	52.08	27	50.00	10	33.33
*f* _3_	↑	5	10.42	9	16.67	2	6.67
	↓	18	37.50	18	33.33	18	60.00
	RI	35	72.92	52	96.30	25	83.33
*p* _1_	↑	13	27.08	1	1.85	4	13.33
	↓		0.00	1	1.85	1	3.33
	RI	24	50.00	45	83.33	28	93.33
*p* _2_	↑	24	50.00	1	1.85	1	3.33
	↓		0.00	8	14.81	1	3.33
	RI	45	93.75	45	83.33	24	80.00
*p* _3_	↑	3	6.25	2	3.70	1	3.33
	↓		0.00	7	12.96	5	16.67
	RI	8	16.67	35	64.81	28	93.33
*p* _4_	↑	40	83.33	6	11.11	1	3.33
	↓		0.00	13	24.07	1	3.33
	RI	9	18.75	36	66.67	15	50.00
*p* _5_	↑	39	81.25	4	7.41	3	10.00
	↓		0.00	14	25.93	12	40.00

**Table 4 ijerph-19-13669-t004:** Immunogram analysis for the state of the immune system *k*_2_.

Ind.	Cond.	Age Range, Years											
		0–3				4–9				10–17			
		↓ *	%	↑	%	↓	%	↑	%	↓	%	↑	%
	RI	2	25	6	75					22	57.89	6	15.79
*f* _1_	↑					2	16.67			4	10.53	6	15.79
	↓						0.00	10	83.33		0.00		0
	RI	3	37.5	5	62.5	3	25.00		0	22	57.89	5	13.16
*f* _2_	↑						0.00		0	6	15.79	5	13.16
	↓						0.00	9	75		0.00		0
	RI		0				0.00		0		0.00		0
*f* _3_	↑		0				0.00		0		0.00		0
	↓		0	5	62.5		0.00	8	66.67	8	21.05	7	18.42
	RI	1	12.5	7	87.5	6	50.00	6	50		0.00		0
*p* _1_	↑		0		0		0.00		0	38	100.00		0
	↓		0		0		0.00		0		0.00		0
	RI	2	25	6	75	6	50.00	6	50	16	42.11		0
*p* _2_	↑		0		0		0.00		0		0.00		0
	↓		0		0		0.00		0	14	36.84	8	21.05
	RI	0	0	0	0	6	50.00	6	50		0.00		0
*p* _3_	↑		0		0		0.00		0		0.00		0
	↓		0		0		0.00		0	26	68.42	12	31.58
	RI	1	12.5	7	87.5	6	50.00	6	50	12	31.58		0
*p* _4_	↑		0		0		0.00		0		0.00	14	36.84
	↓		0		0		0.00		0	12	31.58		0
	RI	1	12.5	2	25	6	50.00	6	50	13	34.21		0
*p* _5_	↑		0	5	62.5		0.00		0		0.00	11	28.95
	↓		0		0		0.00		0	14	36.84		0

* The arrows in the table header show the deviation in the immunoglobulins to the smaller ↓ or more ↑ region relative to the reference interval.

**Table 5 ijerph-19-13669-t005:** Immunogram analysis for the state of the immune system *k*_3_.

Ind.	Cond.	Age Range, Years											
		0–3				4–9				10–17			
		↓ *	%	↑	%	↓	%	↑	%	↓	%	↑	%
	RI	1	50.00	1			0.00	2	33.33		0.00		0.00
*f* _1_	↑				0.00	2	33.33	2	33.33		0.00	2	22.22
	↓		0.00		0.00		0.00		0.00	7	77.78		0.00
	RI	1		1	12.50		0.00	2	33.33		0.00		0.00
*f* _2_	↑		0.00		0.00	4	66.67		0.00		0.00	2	22.22
	↓		0.00		0.00		0.00		0.00	7	77.78		0.00
	RI	1		1	12.50	3	50.00	3	50.00		0.00		0.00
*f* _3_	↑		0.00		0.00		0.00		0.00		0.00	3	33.33
	↓		0.00		0.00		0.00		0.00	6	66.67		0.00
	RI	1		1	12.50	3	50.00	3	50.00		0.00		0.00
*p* _1_	↑		0.00		0.00		0.00		0.00	7	77.78	2	22.22
	↓		0.00		0.00		0.00		0.00		0.00		0.00
	RI	1		1	12.50	3	50.00	3	50.00	6	66.67	3	33.33
*p* _2_	↑		0.00		0.00		0.00		0.00		0.00		0.00
	↓		0.00		0.00		0.00		0.00		0.00		0.00
	RI	1		1	12.50	3	50.00	3	50.00	5	55.56	24	266.67
*p* _3_	↑		0.00		0.00		0.00		0.00		0.00		0.00
	↓		0.00		0.00		0.00		0.00		0.00		0.00
	RI	1		1	12.50		0.00	3	50.00		0.00	3	33.33
*p* _4_	↑		0.00		0.00	2	33.33	1	16.67	6	66.67		0.00
	↓		0.00		0.00		0.00		0.00		0.00		0.00
	RI	1		1	12.50	1	16.67	3	50.00		0.00	2	22.22
*p* _5_	↑		0.00		0.00	2	33.33		0.00		0.00		0.00
	↓		0.00		0.00		0.00		0.00	7	77.78		0.00

* The arrows in the table header show the deviation in the immunoglobulins to the smaller ↓ or more ↑ region relative to the reference interval.

**Table 6 ijerph-19-13669-t006:** Immunogram analysis for the state of the immune system *k*_4_.

Ind.	Cond.	Age Range, Years											
		0–3				4–9				10–17			
		↓ *	%	↑	%	↓	%	↑	%	↓	%	↑	%
	RI					2	25.00		0	2	40.00	3	60
*f* _1_	↑			2	25	1	12.50		0		0.00		0
	↓	6	75		0	4	50.00	1	12.5		0.00		0
	RI	2	25	1	12.5	4	50.00	1	12.5	2	40.00	3	60
*f* _2_	↑	1	12.5		0		0.00		0		0.00		0
	↓	2	25	2	25	3	37.50		0		0.00		0
	RI	4	50	4	50		0.00	2	25	1	20.00		0
*f* _3_	↑		0		0	4	50.00		0		0.00		0
	↓		0		0	2	25.00		0		0.00	4	80
	RI	4	50	4	50	4	50.00	1	12.5	1	20.00	4	80
*p* _1_	↑		0		0	3	37.50		0		0.00		0
	↓		0		0		0.00		0		0.00		0
	RI	4	50	4	50	4	50.00	1	12.5	1	20.00	4	80
*p* _2_	↑		0		0	3	37.50		0		0.00		0
	↓		0		0		0.00		0		0.00		0
	RI	4	50	4	50		0.00		0	1	20.00	4	80
*p* _3_	↑		0		0	4	50.00	1	12.5		0.00		0
	↓		0		0	3	37.50		0		0.00		0
	RI		0		0	6	75.00	2	25	1	20.00	4	80
*p* _4_	↑	4	50	4	50		0.00		0		0.00		0
	↓		0		0		0.00		0		0.00		0
	RI	4	50	4	50	6	75.00	2	25	1	20.00	4	80
*p* _5_	↑		0		0		0.00		0		0.00		0
	↓		0		0		0.00		0		0.00		0

* The arrows in the table header show the deviation in the immunoglobulins to the smaller ↓ or more ↑ region relative to the reference interval.

## Data Availability

Immunograms of studies are stored in the database of the medical centre medical institution “Philosophy of beauty and health”, 64, KIM St., Perm 614990, Russia.

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
