# Peer review of "Impact of the New Coronavirus Infection on the Immune System of Children and Adolescents in the Region of the Russian Federation"

_ijerph, 2022, doi:10.3390/ijerph192013669_

Round 1

Reviewer 1 Report (New Reviewer)

Reviewer’s comments:

In the introduction part, literature reviewing is not done properly, for example in the following lines authors just cited the literature mentioning “discussed/studied/explored” without proper explanation, these  lines should be rewritten.

Perspectives on immune system modulation in the SARS-COV-2 infection using India as an example are discussed in [13]. The relevance of susceptibility of the child immune system to SARS-CoV-2 is explored in the article [14]. The prevalence of the new corona- virus infection among Russian children has been studied in [15]. The status of cellular immunity in children with recurrent respiratory disease has been described in [16]. The problem of differential diagnosis of immunity to SARS-COV-2 was considered at a conference in Sochi in 2021 [17]. The formation of collective immunity to SARS-COV-2 among the population of the Republic of Belarus was studied in [18]. Markers of long-term immunity have been studied in [19].

In the materials and method section, please add a line after “During the pandemic of 2020-2022, 247 people were tested for 113 the new SARS-CoV-2 coronavirus infection” to clearly indicate that these 247 samples were used in the study.

The methods should be described properly with all the steps followed and reagent used, for example the details ELISA, flocytometry, serological tests etc

In the result section line 212- ‘experiments were carried out at the medical institution “Philosophy of Beauty 212 and Health” (Perm) in the period 2020-2021’ but in other places it is mentioned 2020-2022.

In the section Institutional Review Board Statement, the protocol approval number is: protocol code 01.1/14-34-1032, 392 26.08.2022.

The protocol is approved in August, 2022 and the samples were collected during 2020-2022.

Does that mean that the authors collected the samples without any ethical approval? Please justify.

Author Response

Reviewer 2 Report (New Reviewer)

Comments

Kostarev et al presented/analyzed the immunogram/health data of young and adolescents patients Covid-19 patients for the period of  2020-2021 in Russian Federation. They hypothesized that Sars-CoV2 is capable of changing the human immune system by the new wave of coronavirus infection. Although the study of immunological changes in Covid-19 based on strains and/or period is important, but the author represented the data with complicated manner that readers would have difficulty in understanding. Before accepting for publications they should correct the manuscript thoroughly. Some of the examples are:

1.       In abstract and elsewhere, authors mentioned the period of Covid-19 patients studied is 2020-2021. But in materials and methods it is 2020-2022.  The omicron variant (they mentioned but not clear whether they are included) appeared in between  late 2021 and early 2022.

2.       In all tables, the patients are subdivided into three groups and disease state ( k1,k2, k3, k4) but they did not show/devide the data based on each period. Thus, how the wave or period specific immunological modifications are concluded? Instead it shows the differences in age specific manner!!

3.       In all tables, the data representation is complex. For example in table 2, 0-3 age has only numbers but in other age group it is number and percent!! If authors wants to put both, they should write as number(percentages) like  54(72.47%)

4.       In figures, the lines graphs red, blue and green are not defined in the figure legends as which group they represent. Thus, the interpretation of the data is impossible to read.

5.       Table or table legends should include more explanations about data representation (may be with more column). Just arrows are not enough.

6.       In Table 2, the table heading and table legend are almost similar. No point of duplicating them.

7.       Page 6, In line 220, k1 was 132 pc……… pc should be defined first time as percent.

8.       In the model development, all mathematical terms should be more defined or explained, such as line 167, P, F, R10, R20 etc.

Author Response

This manuscript is a resubmission of an earlier submission. The following is a list of the peer review reports and author responses from that submission.

Round 1

Reviewer 1 Report

Dear Authors,

Article found relevant to current scenario, therefore, I hereby wish to appreciate your intellectuals very much for the scientific and informative compilation. But there are few suggestions for the improvement of article:

Comment 1: Need to reframe the title and cut short by reducing the word repetition (Pg 1, line 1-4).

Comment 2: Need to cut short and reduce abstract. No need to repeat entire results in the abstract section. Need to state the objectives & scope, and add elementary Methodology with conclusive results in abstract (Pg 1, line 14-46).

Comment 3: If the heading is only results. Then no need to discuss the finding with relevant citation in the same (Pg 3, line 101-114). Either create separate heading of Discussion to discuss the current findings with earlier published reports or researches or recreate a common heading i.e. "Results and Discussion"

Comment 4: Need to write Results in past tense. Therefore, you are suggested to please check the tense and spelling wherever applicable across the entire article for language and English improvement.

Comment 5: Mention the level of significant during biostatistical data presentation and validation if applicable.

Reviewer 2 Report

Kostarev and colleagues managed to stratify immunograms based on age and disease stage following SARS-Cov2 infection. Presentation from the introduction was not perfect. Gaps are left for the reviewer to fill. I advise that they quote the finding from those papers and then cite. The paper was also very hard to read as there are a lot of variables (f p k) symbols which are not explained well enough etc. For instance, what does to the right of reference mean? Also smaller and more does not sound very scientific. My suggestion will be to use visuals if possible to disseminate findings better, if the data is much, heat maps may assist better too. The tables are too many and heavily rely on the reader having to go back and forth multiple times. The statement on ethics is also missing. This study is very important and warrants attention to be given to presentation as it was lacking.

Reviewer 3 Report

In this article, the authors have studied the changes in the immune system during COVID-19 pandemic among children and adolescents. For this, they have used a study sample size of 247 and FACS as an immunological tool for analysis. The study looks interesting however it needs revisions.

  1. There are no details about the sample collection place, ethical approval, or getting the patient's consent. Even though the authors have an informed consent statement there should be a statement about ethical approval inside the text.
  2.  Introduction: Line 52 mentions, "Metchnikov as the founder of immunology". It will be more appropriate if they say as Father of Cellular Immunology (DOI: 10.3389/fimmu.2012.00068).
  3.  Many sentences in the introduction are incomplete. For example line 60  "was studied in".
  4. The methodology part was not clear. How the serum was processed for FACS was not described.
  5. What kind of COVID vaccine is given to the subjects in the study? The study was conducted before or after the COVID vaccination?
  6. The authors need to rewrite the entire article for better understanding. Many statements are confusing.